**communications** engineering

# Percutaneous nephrostomy guidance by a convolutional-neural-network-based optical coherence tomography endoscope
Chen Wang [1], Paul Calle [2], Feng Yan[1], Qinghao Zhang[1], Kar-Ming Fung[3], Zhongxin Yu[3], Sean G. Duguay[4], William B. Vanlandingham[4], Nathan A. Bradley[5], Sanjay G. Patel[5], Bradon Nave[6], Clint Hostetler[6], Ashley Milam[6], Chongle Pan [2] ✉ & Qinggong Tang [1] ✉

Percutaneous nephrostomy is widely used in kidney access surgeries. Despite its prevalence in urological interventions, it presents two operational challenges: 1) precise needle placement into the renal pelvis; and 2) avoiding hemorrhage from blood vessel rupture. In this study, we developed an endoscopic optical coherence tomography probe for needle navigation. We conducted experiments on thirty-one human kidneys for two aspects: 1) tissue recognition, and 2) blood vessel detection. Experimental results indicated that renal tissues including cortex, medulla, calyx, sinus fat, and pelvis could be effectively distinguished through structural optical coherence tomography imaging, and renal blood flow could be detected through the Doppler function. Deep learning methods were utilized to automate recognition procedures. For tissue classification, an Inception model was used, achieving a recognition accuracy of 99.6%. For blood vessel detection, an nnU-net model was applied, exhibiting an intersection over union value of 0.8917 for blood vessel and 0.9916 for background.

Urolithiasis is one of the most common diseases which affects the urinary tract, with renal stones being a frequent manifestation. Over half a century, the prevalence of renal stones has exhibited a noteworthy increase: from merely 3.8% in the late 1970s, to 8.8% in the late 2000s, and has reached up to 11.0% in recent years[1,2]. Urolithiasis treatment requires surgical procedures such as urine drainage, stone breaking (lithotripsy), stone removal (percutaneous nephrolithotomy, PCNL), and other therapeutic interventions[3]. Nevertheless, conventional access through the ureter is not feasible when stone obstructions or urinary tract infections are present. Percutaneous nephrostomy (PCN) is a minimally invasive technique employed in varied renal surgical interventions[4]. It involves the needle insertion through the patient's back or flank to establish a direct pathway into the renal pelvis, allowing for urinary drainage and subsequent surgical interventions[5]. Precise selection of an insertion site is crucial for PCN success and patient safety. An ideal needle track needs to be short and straight, ensuring minimal tissue damage before arriving at the renal collecting system[6]. Needle misplacement increases the risk of injuries and complications, primarily attributed to two main factors: (1) injury to kidney tissue, and (2) rupture of renal blood vessels. Tissue damage due to needle misplacement leads to operational challenges such as kidney displacement, collapse of the collecting system,

and extravasation of renal stone fragments[6]. Vascular damage, particularly to the segmental renal artery, can precipitate haemorrhage, fistula formation, or segmental infarction. Furthermore, minor temporary postoperative bleeding has been reported in approximately 95% of all PCN cases[7]. An advanced medical imaging method is required to improve the safety and efficacy of PCN.

Currently, ultrasound and fluoroscopy have been widely utilized for PCN navigation. Ultrasound is applied for its safety and convenience, but it has variable effectiveness due to its limited resolution. It has been reported that the ultrasound-guided single-puncture success rates ranged from 34.3% to 91.2%[8–11]. Fluoroscopy has superior imaging contrast. Similar to ultrasound, however, its limited resolution leads to single-puncture success rates ranging from 50.0% to 79.9%[11–13]. These modalities can provide a general view of the kidney with PCN needle tract, but lack sufficient detailed information for precise needle localization.

Optical coherence tomography (OCT) is a well-established medical imaging modality with superior axial resolution at the micrometer level, approximately ten times higher than conventional imaging techniques like ultrasound and fluoroscopy[14]. A traditional endoscope has been reported to realize PCN needle-tip visualization, but it does not provide depth-resolved

[1]Stephenson School of Biomedical engineering, University of Oklahoma, Norman, OK, 73019, USA. [2]School of Computer Science, University of Oklahoma, Norman, OK, 73019, USA. [3]Department of Pathology, University of Oklahoma Health Sciences Center, Oklahoma City, OK, 73104, USA. [4]Department of radiological sciences, University of Oklahoma Health Sciences Center, Oklahoma City, OK, 73104, USA. [5]Department of Urology, University of Oklahoma Health Sciences Center, Oklahoma City, OK, 73104, USA. [6]LifeShare Network, Inc, Oklahoma City, OK, 73132, USA. ✉e-mail: cpan@ou.edu; qtang@ou.edu

tissue information. OCT is capable of being integrated with endoscopes[15]. In this study, we developed a forward-view endoscopic OCT probe system integrated with the PCN needle (Fig. 1). It can acquire depth-resolved tissue information, potentially for precise tissue recognition at the PCN needle tip. In addition, OCT detects blood flow by Doppler mode[16]. Therefore, we applied our endoscopic OCT to assist PCN navigation, with (1) structural imaging function for tissue recognition, and (2) Doppler OCT function for blood vessel detection. Since extensive training of urologists and interventional radiologists to use OCT takes time and effort, we proposed to use computer-aided methods to automate the post-procedure imaging data analysis. Convolutional neural network (CNN) has been employed in medical imaging research. Clinically, it has been applied in intraoperative tissue assessment and blood vessel segmentation[17,18]. To assist endoscopic OCT imaging, our previous studies have illustrated the potential of using CNN methods in different applications, such as tumor evaluation, pig kidney tissue classification, epidural needle navigation, Veress needle location[19–23], etc. Furthermore, CNN has been applied in the image segmentation task[24]. In this study, we utilized the following CNN architectures to fulfill the two objectives: (1) ResNet50, Inception, and Xception for renal tissue recognition; and (2) an automatic pipeline, nnU-Net, for blood vessel segmentation. A complete CNN-aided endoscopic OCT imaging platform was established to assist PCN needle navigation.

## Methods

### Endoscopic OCT system setup

To realize the depth-resolved imaging of kidney tissue at the PCN needle, we established an endoscopic OCT system by using a gradient-index (GRIN) rod lens, as shown in Fig. 1a. In the perpendicular direction to the optical axis of the GRIN lens, the refractive index decreases from the center to the edge to keep the light propagating inside it[25]. It transmits the imaging information from one end to the other without loss of spatial resolution. Typically, PCN needles used have a gauge size of 16 or 18[7,26]. In our system, we utilized the GRIN rod lens with a diameter of 1.30 mm and a length of 138.60 mm to fit the hollow bore of PCN needle. Our system uses a swept-source laser source with an output power of 25 mW, a central wavelength of 1300 nm, and a tuning bandwidth of 100 nm. The axial-scan rate of the OCT system can reach 200 kHz, enabling high-speed real-time imaging. The output laser power was split to a 3%: 97% ratio by a fiber coupler (FC). The 3% portion was used to trigger OCT sampling through a Mach–Zehnder interferometer (MZI), and the remaining 97% was directed to an optical fiber circulator for OCT imaging. The circulator ensured unidirectional light transmission, directing the imaging laser from port 1 to exit through port 2, and subsequently into another FC where the laser is split

equally to the sample and reference arms, therefore establishing a Michelson interferometer setup. In each arm, a polarization controller was used to reduce noise during OCT imaging. Within the sample arm, the light was directed for imaging the renal tissue using a galvanometer scanning mirror (GSM), through a sample arm GRIN lens integrated in front of the scanning objective lens of the GSM. The proximal end surface of the GRIN lens was positioned at ~0.3 mm from the intermediate image plane of the GSM scanning objective lens to mitigate the strong reflection from the GRIN lens surface. The backscattered tissue signals returned to the FC for post-processing. In the reference arm, a collimator was integrated to transmit the light from the optical fiber to space. A convex lens was employed to converge the light into the GRIN lens. After entering the space, the light passed through another GRIN lens (similar to the one in the sample arm), which served to expand the optical length and compensate for light dispersion from the sample arm GRIN lens. Then, the light was reflected by a mirror and retraced its path back through the original route to the same FC where the sample arm light returned. The two returned lights combined to be an interference signal. It was initially separated by the FC and circulator, and then transmitted to a balanced detector (BD), which was utilized for common-mode noise cancellation. Finally, the BD output signal propagated to a data acquisition (DAQ) board of the computer for processing and monitoring. This system presented the highest scanning speed at 200 kHz (A-Scan), an axial resolution of 11 μm, a lateral scanning resolution of 20 μm, and a sensitivity of 92 dB. A GRIN lens compatible with the PCN needle will be put into the hollow space within the PCN needle as illustrated in Fig. 1b. In clinical practice, urologists and interventional radiologists can concurrently insert the PCN needle with the GRIN lens inside together into the kidney. Ultrasound provides an overall imaging of the entire kidney area, and our endoscopic OCT system offers a high-resolution visualization of the front view of the PCN needle tip. Real-time OCT imaging at the needle tip can be realized using a high-speed scanner. Once the PCN needle reaches the target area (renal pelvis), the GRIN lens can be retracted with the scanner, leaving the PCN needle in place for subsequent surgical interventions. It is worth noting that our probe does not introduce additional injuries or require extra time, thereby ensuring safe and efficient PCN needle navigation.

### Data acquisition of human kidneys and CNN-based prediction

This research was approved by the University of Oklahoma and the University of Oklahoma Health Sciences Center Institutional Review Board (IRB) (Study number: IRB #12462). Organs are provided by Lifeshare of Oklahoma. All samples underwent preservation through hypothermic machine perfusion (HMP) to maintain their physiological status. All

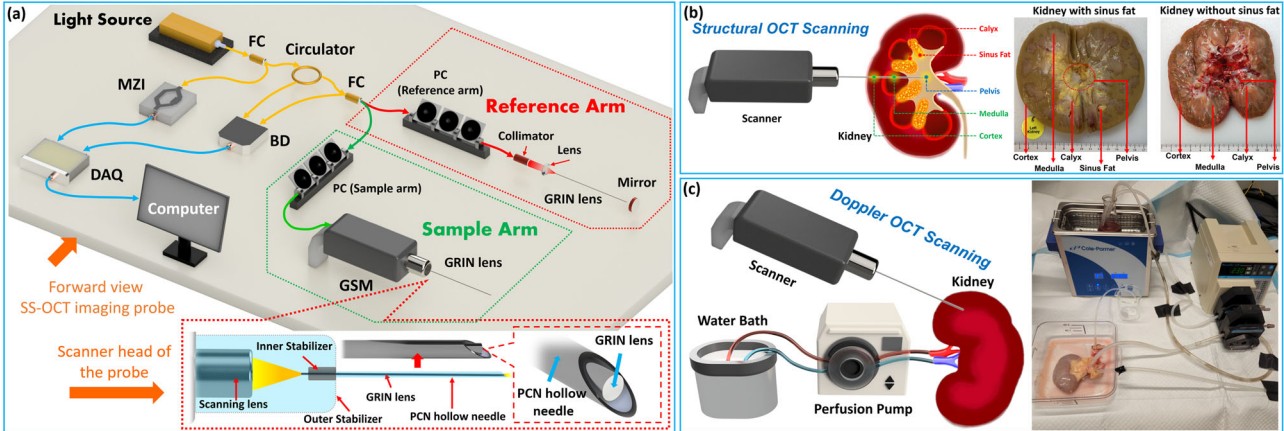

**Fig. 1 | Experimental setup and procedures of the endoscopic OCT imaging.**
**a** Schematic of the forward-view endoscopic OCT imaging probe. FC fiber coupler, MZI Mach–Zehnder interferometer, GSM galvanometer scanning mirror, GRIN lens gradient-index lens, BD balanced detector, DAQ data acquisition. **b** Different

renal tissue imaging by using structural OCT scanning and representative human kidneys with and without renal sinus fat. **c** Schematic of the perfusion pump system for Doppler OCT imaging of renal blood vessels and the experimental setup of the perfusion system.

experiments were conducted on ex vivo human kidneys. Kidneys were imaged within 2 days after removal from donors. We utilized 31 human kidneys in total. Twenty-four were used to establish the dataset (17 for tissue recognition; 5 for blood vessel detection), and 7 kidneys were utilized for blind tests (5 for tissue recognition; 2 for blood vessel detection).

For the tissue recognition task, all the kidneys were dissected to expose their mid-pole. Different tissue types could be visually distinguished based on their colors, shapes and positions. In the experiments, we placed our probe tip against these tissues and acquired their structural OCT images separately. We obtained cross-sectional OCT images with a size of $650 \times 1050$ pixels (the scanning schematic is shown in Fig. S1 of Supplementary Document). It should be noted that renal sinus fat does not always exist, as shown in Fig. 1b. It was absent in two of the utilized kidneys. Our study aimed to distinguish renal tissues through structural OCT imaging. In a typical PCN procedure, the PCN needle should sequentially pass through the renal cortex, medulla pyramid, through the renal papilla at the apex of the calyx, and eventually reach the renal pelvis. If a deviating puncture happens, the PCN needle could pass through the renal sinus fat located between adjacent medulla pyramids and then lead to tearing of the calyx walls. Cortex, medulla, calyx, sinus fat, and pelvis were included in our imaging dataset. We used our OCT endoscope to scan these tissues at the PCN needle tip to help ascertain whether the insertion path is risky. Each of the five tissue layers is characterized by a specific anatomical location and histological structure. Cortex is located at the outermost layer of the kidney, and it is composed of renal corpuscles, tubules and cortical interstitial tissue. The renal medulla, located beneath the outer cortex, consists primarily of loops of Henle, collecting ducts and vasa recta. The renal calyx is located at the innermost part of the kidneys, surrounding the renal pyramids formed by medulla tissue, and it is composed of transitional epithelium and smooth muscle. Renal sinus fat is situated between adjacent renal pyramids and comprises adipocytes and connective tissues. The pelvis is positioned toward the central medial side of the kidney and serves as the cavity that collects the urine from the calyces, so it appears as an empty or fluid-filled space during imaging.

For the blood vessel detection task, we conducted experiments using human kidneys under perfusion to mimic practical conditions (Fig. 1c). The renal artery and vein were cannulated and perfused using a pulsatile perfusion pump. The perfusion solution included red blood cells (RBC) and Ringer's solution[27]. The solution flow rate was set within the range of 100–250 ml/min for normothermic ex vivo kidney perfusion (NEVKP)[28]. A water bath heater was utilized to maintain the temperature at a constant 37 °C. Artificial urine is infused into the renal pelvis via the ureter before the experiment to mimic a practical situation. Once a blood vessel is detected in front of the probe tip, we would slowly move the probe towards the vessel and then stop it prior to penetrating the blood vessel. Doppler OCT images (with a size of $350 \times 450$ pixels) were captured at different distances between the OCT probe tip and the Doppler OCT signal edges, which corresponded to the renal vasculature walls.

## CNN-aided classification task, segmentation task, and blind test

To improve working efficiency and reduce the working burden of doctors, we proposed to use deep learning methods to automate the tissue recognition and blood vessel detection procedures. The CNN architecture was chosen for image classification and segmentation tasks. A description of our data acquisition and testing is shown in Fig. 2a. Our dataset followed the hierarchy of: kidney → scan sequence/trajectory → frames. For each tissue in each kidney in the tissue recognition task, 10,000 images were acquired for dataset establishment, and 1000 images for blind tests. For the blood vessel detection task, we collected 5000 Doppler OCT images from 7 kidneys for dataset establishment, and 100 additional Doppler OCT images for blind tests.

For the tissue classification task, we selected three CNN architectures: ResNet50, Inception, and Xception, all of which have been applied in medical imaging classification tasks across different domains, such as pneumonia X-ray analysis[29], breast cancer histopathology examination[30], and diabetic retinopathy identification[31]. A 17-fold dataset was applied for model selection. As illustrated in Fig. 2b, we constructed our dataset using 12 kidneys for a repeated holdout method. Among them, we randomly selected one kidney to assess the validation prediction. We repeated this procedure 5 times and identified the best-performing CNN architecture as the final one. Then, the remaining five kidneys were utilized for internal testing. This data construction was designed to ensure an effective evaluation framework that can efficiently demonstrate the performance of our models in clinical settings. Specifically, it estimates the performance for new kidneys that were not previously encountered during training. 10,000 OCT images for each tissue type across each kidney were utilized in the model training. The

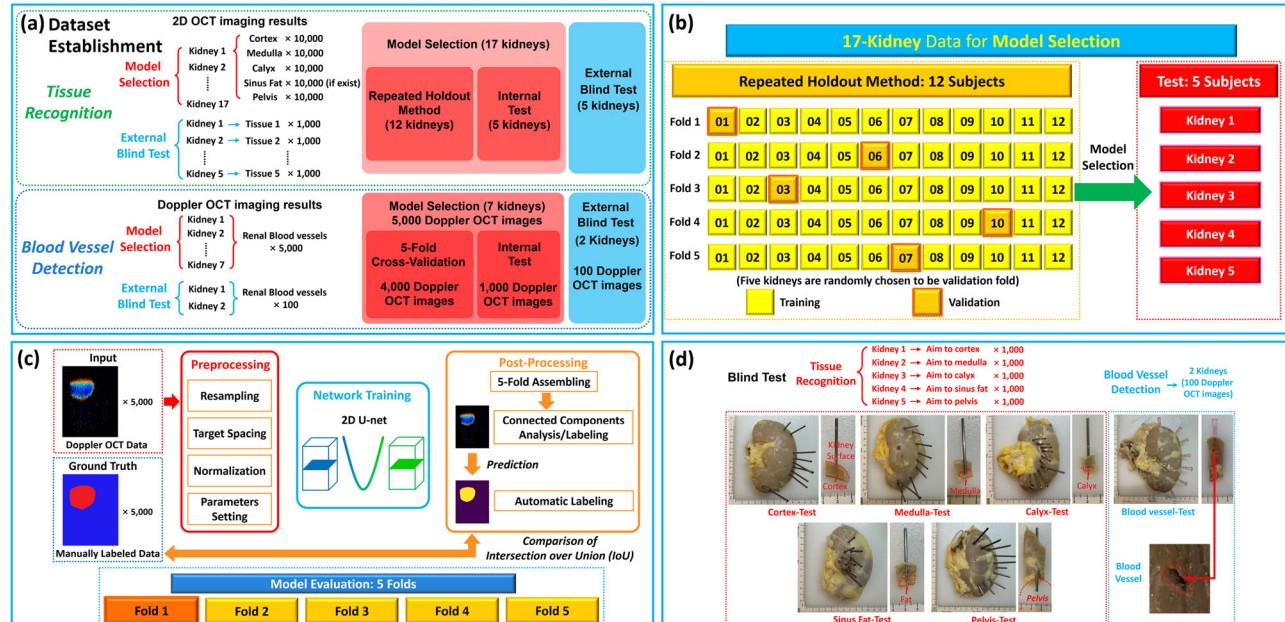

**Fig. 2 | Kidney tissue identification and blood vessel detection procedures using CNN. a** Data acquisition of tissue recognition and blood vessel detection. **b** Model selection by repeated holdouts for renal tissue recognition using CNN. **c** Procedure of using nnU-Net for renal blood vessel segmentation. **d** Data acquisition for external blind tests and pictures of the kidneys under blind tests.

recognition performance was assessed by accuracy, recall and precision, which are defined as follows:

$$Accuracy = (TP + TN)/(TP + FP + TN + FN) \tag{1}$$

$$Recall = TP/(TP + FN) \tag{2}$$

$$Precision = TP/(TP + FP) \tag{3}$$

where TP represents true positive, TN represents true negative, FP represents false positive, and FN represents false negative. Accuracy demonstrates an overall classification level of a deep learning model. Recall presents the ability of the deep learning model to predict the target category from all the objects. Precision shows how many times the deep learning model correctly predicts the target category.

For the blood vessel segmentation task, we selected U-Net model, which has gained widespread recognition for its effectiveness in various biomedical image segmentation tasks[32]. The nnU-Net, representing an enhanced iteration of the original U-net framework, was chosen for our blood vessel segmentation[24]. Specifically, we employed the 2D U-net configuration option within the nnU-Net framework, as detailed in Fig. 2c. Among the 5000-Doppler imags, 4000 were used for 5-fold cross-validation and the other 1000 were used for the internal test. For the prediction of test results, the library used an ensemble model composed of the five models generated from the five-fold cross-validation. We set the model with 1000 epochs, and the stochastic gradient descent with a high initial learning rate of 0.01 and a large Nesterov momentum of 0.99. Intersection over Union (IoU) metric was adopted as the primary measure for evaluating segmentation performance. It indicates the overlap between the predicted segmentation and the ground truth, shown as a ratio of their intersection over their union. IoU provides a quantitative assessment of the model's ability to accurately delineate the area of blood vessels within the images, comparing the overlap between the model's predictions and the manually labeled gold standard. The formula for IoU is:

$$IoU = (Area\ of\ Overlap)/(Area\ of\ Union) \tag{4}$$

and its value ranges from 0 to 1, where 0 indicates no overlap and 1 indicates perfect overlap. A higher value of IoU represents a better performance of the segmentation model. This metric is instrumental in gauging the precision of the segmentation process, highlighting the effectiveness of the nnU-Net model in distinguishing blood vessel structures from the surrounding renal tissues in Doppler OCT images.

To validate the practical applicability, we further conducted blind external tests on additional unseen kidneys. Before the experiments, we equipped stainless-steel tubes and clinically used needles as positional markers on our OCT endoscope for tissue recognition and blood vessel detection, separately. After the experiments, the markers remained at the original place, thereby serving as reference points for the locations of the renal tissues and blood vessels that were scanned, as shown in Fig. 2d.

For tissue recognition, we first inserted our probe into the kidney and started the OCT scanning. When the OCT imaging result indicated the imaging features of a specific tissue type, we stopped insertion and recorded the results. Subsequently, the OCT probe was withdrawn, leaving the stainless-steel tube marker in its original position to denote the OCT imaging location. This procedure was replicated across all five kidneys for each

of the tissue types under investigation. We placed our endoscopic OCT probe on the cortex for kidney No. 1, the medulla for kidney No. 2, the calyx for kidney No. 3, the sinus fat for kidney No. 4, and the pelvis for kidney No. 5, respectively. Upon completion of these tests, the imaging sites, along with the markers, were exposed by cutting the kidney, and the type of renal tissue at each location was determined through visual examination. Blind test prediction was conducted based on the well-trained CNN model to further validate its performance.

For blood vessel detection, ten insertions were executed on each of the extra two unseen perfused kidneys. The insertion was paused upon detection of Doppler signals, allowing for the recording of the renal blood vessel imaging results and leaving the needle in position to mark the site. The regions that indicated a blood vessel were subsequently sectioned. Renal blood vessels were examined at all needle insertion sites, as shown in Fig. 2d, to validate the prediction performance.

### Control experiments of using ultrasound for PCN navigation
Ultrasound is the most commonly used imaging technique during PCN procedure. To further validate the improvement by our system, we performed control experiments using structural ultrasound for tissue recognition, and Doppler ultrasound for blood vessel detection. In the structural ultrasound experiments, we blindly inserted a clinically used 16-gauge needle into the human kidneys and conducted the ultrasound scanning. The needle tip arrived at different renal tissue types and the needles were left in the kidneys. Then we cut the kidneys to verify the tissue types at the needle tip locations. At the same time, we gave the corresponding structural ultrasound imaging results to two professional radiologists for tissue type recognition. We conducted a total of 50 insertions from five additional kidneys and compared the tissue recognition rates of the radiologists against the actual tissue types. This analysis aimed to evaluate the precision of structure ultrasound in identifying renal tissue types at the needle tip. Furthermore, we conducted Doppler ultrasound imaging of human kidneys under perfusion situations. This approach visualized the renal blood flow, and we would compare them with our CNN-aided endoscopic Doppler OCT imaging.

### Reporting summary
Further information on research design is available in the Nature Portfolio Reporting Summary linked to this article.

## Results
To realize real-time feedback during PCN navigation, a high-performance server was employed to accelerate data processing within the CNN framework. In this study, we used a Linux server with Ubuntu 20.04.6 LTS with two NVIDIA RTX A6000 GPUs for all CNN tasks. For tissue recognition using a batch size of 32 images, it required an average prediction time of 6.07 ms per image. For blood vessel segmentation, the average processing time was 250 ms. These results affirmed the real-time performance of our OCT imaging platform, illustrating its suitability for clinical applications where timely decision-making is critical. By utilizing our technique to reduce tissue damage and blood vessel injury, patients under PCN can expect improved safety and reduced medical costs. Basic information of the GPU and prediction performance are listed in Table 1.

### Renal tissue recognition results
Figure 3a demonstrated the OCT structural imaging results of each renal tissue type. We first obtained two-dimensional (2D) cross-sectional images

**Table 1 | Data processing hardware and prediction performance**

| GPU information | | | | | Prediction performance | |
|---|---|---|---|---|---|---|
| GPU model | Memory | Memory interface | Memory bandwidth | CUDA cores | Tissue recognition time/per image | Blood vessel segmentation time/per image |
| NVIDIA RTX A6000 | 48 GB | 384-bit | 768 GB/s | 10,752 | 6.07 ms | 250 ms |

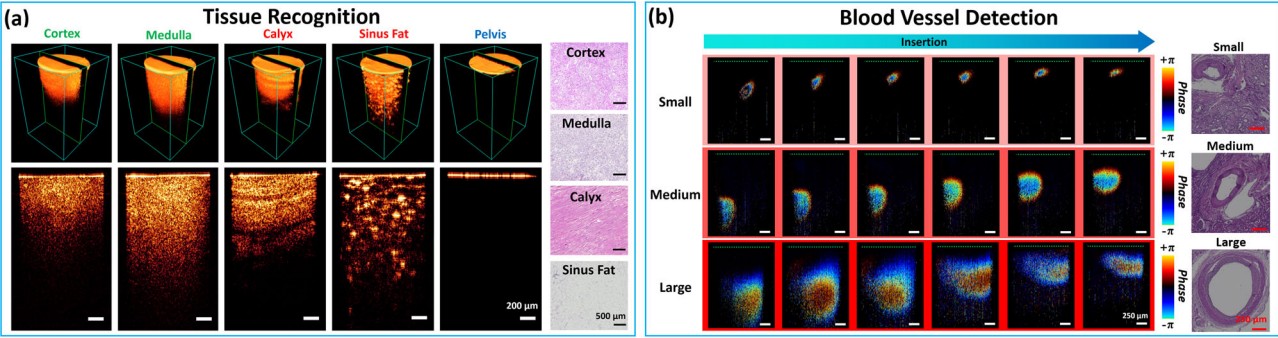

**Fig. 3 | Endoscopic imaging results. a** Structural OCT imaging results of different renal tissue types and corresponding histology results of different renal tissue types. **b** Doppler imaging results of renal blood vessels with three representative different sizes and their corresponding histology results.

with a size of 1.3 mm × 2.1 mm (X by Z). Adjacent 2D cross-sectional images were combined to reconstruct three-dimensional (3D) images with a volume of 1.3 mm × 1.3 mm × 2.1 mm. This scanning area was limited by the diameter of the GRIN lens rather than the field of view of the GSM (10.0 mm × 10.0 mm). We further acquired histology results of different renal tissues, which showed different biological features. It is worth noting that the pelvis is the cavity that collects the urine, so there was no histological result of it.

The OCT imaging results indicated that these human renal tissues possessed different imaging characteristics. The cortex and medulla presented relatively homogeneous distributions of pixel values in the tissue areas, with a gradual overall decrease in the vertical direction. Notably, the imaging depth of the medulla was much larger than that of the cortex. Calyx tissue, comprising mucous membranes overlaid with connective tissue, manifests loose-to-dense biological features, resulting in transverse dark-bright stripes in the OCT images based on intensity changes. Renal sinus fat, composed of adipose tissue containing large globules of sinus fat (lipid droplets) and fiber structural networks, could be distinguished in the OCT images by its bright spots and reticulate patterns. Renal pelvis, serving as a urine collection chamber, was characterized by the absence of tissue signals beneath the probe tip. Its OCT imaging result featured an isolated bright line at the top, representing the GRIN lens surface, and an empty background without tissue signal beneath. From these results, our endoscopic OCT probe shows promise in recognizing different renal tissues.

### Blood vessel detection results

Three groups of representative Doppler OCT imaging results of the renal blood vessels with three different sizes: small, medium, and large, were shown in Fig. 3b. Based on the insertion routes of these blood vessels, we also conducted histological analysis of them. Doppler OCT allows for the differentiation of blood flow velocities across different positions, represented by the phase-based color coding in the images. Blood flow speed can be calculated based on Doppler frequency shift and other parameters as shown below:

$$f_D = 2nV \cdot \cos\alpha / \lambda_0 \tag{5}$$

where $f_D$ is the Doppler frequency shift, $n$ is the refractive index of the medium, $\alpha$ is the Doppler angle between the light propagation and the flow velocity direction, and $\lambda_0$ is the central wavelength of the OCT laser source. The phase shift is limited to a range of $-\pi$ to $+\pi$ mathematically, meaning that Doppler OCT can only accurately measure flow velocities within a specific range.

GRIN lens surface was not displayed under Doppler OCT mode, so we labeled them based on the corresponding structural OCT results with green dashed lines on the top. The demonstration of our OCT probe approaching the renal blood vessels was illustrated within each group. In the experiments, we simulated the situation in which our probe approached the renal blood

vessel in front. The Doppler OCT images across the different groups revealed that as the probe surface approached and provided pressure on a renal blood vessel, the distance between the probe tip and blood vessel decreased, and the cross-sectional appearance of the blood vessel became increasingly flattened. These results demonstrated our endoscopic OCT probe's efficacy in renal blood vessel detection. In addition, renal blood vessels with a diameter of less than 200 μm could be clearly identified, as illustrated in the "small" group of Fig. 3b. By accurately mapping out the sizes and proximities of these small blood vessels relative to the needle tip, our OCT probe offers a significant advantage in preoperative and intraoperative planning during PCN. This would assist radiologists in estimating the risk of blood vessel injury and determining subsequent steps to ensure surgical safety and efficiency.

### Control experiment results

Imaging results of the control experiments using ultrasound were presented in Fig. 4. The structural ultrasound images allowed for the needle's overall trajectory visualization. However, it was difficult to recognize the specific tissue type at the needle tip. We conducted a detailed analysis of feedback from the two professional radiologists to assess the efficacy of tissue recognition using ultrasound. The confusion matrices of the five renal tissue predictions were shown in Supplemental Table S1. Out of all the 50 tissue recognitions, radiologists achieved only 14 and 15 correct identifications (28% and 30% recognition rates, respectively). For using Doppler ultrasound in blood vessel detection, the transducer commonly used in PCN typically operates at a frequency of 3.5–5.0 MHz[33]. It is challenging for Doppler ultrasound within this frequency range to determine the presence of small blood vessels with a diameter of less than 1.5 mm[34]. From the imaging results, it is obvious that an overall vessel distribution can be visualized, but it is difficult to accurately evaluate its size and distance to the needle tip.

### CNN prediction of tissue recognition

Three CNN architectures—ResNet50, Inception, and Xception—enabled a comprehensive evaluation of these models' capabilities in classifying renal tissues. The model training was structured around a repeated holdout method involving 12 kidneys. Five repetitions (one kidney for each) were conducted for the validation process, thereby ensuring a robust and unbiased evaluation of their performances. Then, we conducted an evaluation of the best architecture of the trained CNN models using an internal test dataset of the 5 kidneys to demonstrate their performance.

Table 2 illustrates the accuracy values for each of the five repetitions. All three CNN architectures performed outstanding prediction results. Among them, Inception achieved the highest average accuracy of 99.6%, followed by Xception of 99.5%, and ResNet50 of 98.9%. Therefore, we chose Inception as the final prediction model.

Table 3a demonstrated the average confusion matrix of the internal test dataset for the final Inception model. The renal pelvis showed an ideal test

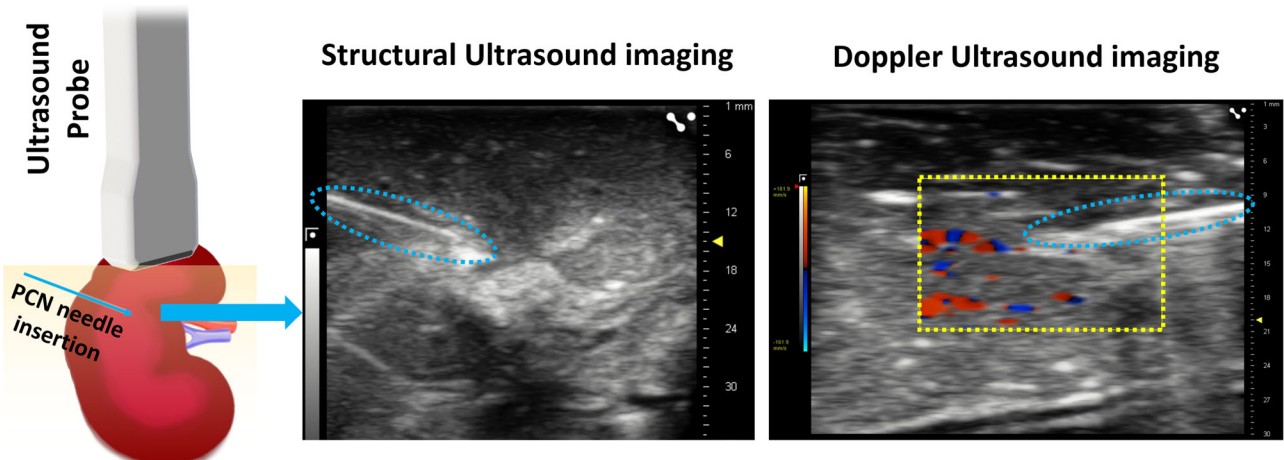

**Fig. 4 |** Control experiment imaging results. Sturctural ultrasound (left) and Dopplerultrasound (right) imaging results of an ex vivo human kidney under perfusion.

**Table 2 | Average tissue prediction values of the repeated holdout method in different CNN architectures**

|  | R1 | R2 | R3 | R4 | R5 | Average | Std error |
|---|---|---|---|---|---|---|---|
| Resnet50 | 99.2% | 99.8% | 98.2% | 98.7% | 98.7% | **98.9%** | **0.27%** |
| Inception | 99.7% | 100.0% | 99.3% | 98.9% | 99.9% | **99.6%** | **0.21%** |
| Xception | 100.0% | 99.9% | 98.6% | 99.0% | 99.9% | **99.5%** | **0.29%** |

The bold values represent the average and standard error, demonstrating the overall performance of each CNN architecture.

**Table 3 | Average tissue prediction results using the final Inception model for the internal test dataset**

|  |  | Prediction | | | | |
|---|---|---|---|---|---|---|
|  |  | Cortex | Medulla | Calyx | Sinus fat | Pelvis |
| (a) Confusion matrix | | | | | | |
| Truth | Cortex | **9989.4 ± 10.6** | 0.0 ± 0.0 | 10.6 ± 10.6 | 0.0 ± 0.0 | 0.0 ± 0.0 |
|  | Medulla | 0.0 ± 0.0 | **9998.0 ± 2.0** | 2.0 ± 2.0 | 0.0 ± 0.0 | 0.0 ± 0.0 |
|  | Calyx | 0.0 ± 0.0 | 0.0 ± 0.0 | **9999.8 ± 0.2** | 0.2 ± 0.2 | 0.0 ± 0.0 |
|  | Sinus Fat | 9.4 ± 9.4 | 0.0 ± 0.0 | 10.8 ± 10.8 | **9979.8 ± 20.2** | 0.0 ± 0.0 |
|  | Pelvis | 0.0 ± 0.0 | 0.0 ± 0.0 | 0.0 ± 0.0 | 0.0 ± 0.0 | **10000.0 ± 0.0** |
| (b) Accuracy, recall, and precision values | | | | | | |
| Avg accuracy | | 99.960% | 99.996% | 99.993% | 99.959% | 100.000% |
| Avg recall | | 99.894% | 99.980% | 99.998% | 99.798% | 100.000% |
| Avg precision | | 99.906% | 100.000% | 99.768% | 99.998% | 100.000% |

The blod values indicate the correct predictions in the confusion matrices, more directly showing the prediction accucacy.

prediction, with no true pelvis predicted to other tissues and no other tissues predicted to the pelvis. This should be attributed to the obvious space under the GRIN lens surface in the OCT results. The calyx also exhibited very promising predictions, with a mere average of 0.2 images predicted as sinus fat. The renal sinus fat exhibited the comparatively lowest performance in prediction, with an average of 9.4 misclassified as cortex and an average of 10.8 misclassified as calyx. Furthermore, on average, 10.6 cortex results and 2.0 medulla results were erroneously predicted as calyx.

Table 3b showed the average prediction performance based on accuracy, recall and precision. The accuracy/recall/precision values on predicting all the tissue types were over 99.7%, which could be considered promising in tissue recognition by using our final Inception model. The micro-average receiver operating characteristic (ROC) curves based on the three CNN architectures among the five tested kidneys are included in Fig. 5, with all the curves superimposed together and the areas under the curve (AUC) reaching 1.00.

## CNN segmentation of blood vessels
The nnU-Net was applied for renal blood vessel segmentations. As illustrated in Fig. 6a, we manually labeled the blood vessel signals in red and the background in blue. These manually labeled results were designated as the ground truth. For nnU-Net prediction, the model outputs 1 for blood vessel signals (marked in yellow) and 0 for background (marked in purple).

Figure 6b shows IoU histograms of blood vessels and background, demonstrating the distribution of the signal identifications of each Doppler imaging result. Horizontal axes represented the IoU value, and the vertical axes represented the amount of the corresponding Doppler OCT images. An optimistic average value of 0.8917 was obtained for renal blood vessel detection, and 0.9916 for background segmentation. These values are promising in medical imaging, indicating that the model is highly accurate in delineating the target blood vessel and background areas from the surrounding imagery. The peak of the IoU histogram distribution was observed at approximately 90%. Analysis of our dataset revealed that 49.4% of all

blood vessel segmentations achieved an IoU exceeding 0.90. Notably, 12.1% of blood vessel predictions exhibited an IoU higher than 0.95, which could be observed in the histogram. For background region segmentation, an impressive 99.3% of all outcomes achieved an IoU value exceeding 0.95. These results indicated exceptional performance of using nnU-Net in automatic renal vascular segmentation in Doppler OCT images.

## Blind test predictions

For tissue recognition, the blind test confusion matrix was shown in Table 4a, and prediction results were shown in Table 4b. Consistent with the training model, the pelvis exhibited ideal predictions with no false positives or true negatives. All accuracy/recall/precision results were over 90%, further proving the promising performances of our CNN model. Cortex and medulla also yielded exceptionally high prediction performances, with accuracy at 99.06%/99.96%, recall at 98.60%/99.80%, and precision at 96.95%/100.00%, respectively. Calyx and sinus fat demonstrated 98.32%/97.42% in accuracy, 91.6%/96.9% in recall, and 100.00%/90.82% in precision, respectively. However, there are slight differences between the internal test and the blind test results. Primarily, calyx, rather than sinus fat, provided the lowest predictive performance. Additionally, all incorrect predictions of the cortex were erroneously classified as sinus fat, diverging from the

expected misclassification as calyx observed in the internal test. Moreover, the entirety of the 0.31% misclassifications of sinus fat were incorrectly predicted to be cortex, with no instances of misclassification as calyx. This discrepancy in predictions, particularly concerning the calyx tissue, could be attributed to the presence of renal sinus fat adjacent to the calyx in kidney anatomy. The kidney samples we utilized for blind tests contained a higher proportion of adipose tissue beside the renal calyx compared to those used in the training model. Consequently, the closer the proximity of calyx and sinus fat, the more likely it is that calyx tissue will be mistakenly recognized as sinus fat in the blind test outcomes. More details of model evaluation are shown in Tables S2 and S3 in the supplementary document.

For blood vessel segmentation, another three representative blind test results of vessels in small, medium, and large sizes are shown in Fig. 7a. Histological processing of the corresponding renal blood vessel imaging areas was conducted. Doppler OCT signals were found at corresponding histological positions. The trained nnU-Net model was utilized for automatic segmentation. We manually labeled the 100 results and considered them as ground truth. The IoU histograms were shown in Fig. 7b, with the high degrees of IoU values for the blood vessel at 0.9074 and background at 0.9670, further affirming the robustness of the renal vascular segmentation within renal tissues using our nnU-Net model.

## Discussion

In this study, we developed a forward-viewing endoscopic OCT probe for PCN needle navigation. This design realizes real-time OCT scanning at the PCN needle tip, without introducing any extra invasiveness. Experiments on human kidneys evaluated its effectiveness in (1) Renal tissue recognition by structural OCT, and (2) Blood vessel detection by Doppler OCT. Five renal tissues show different features in structural OCT results, and blood vessel cross-sectional areas can be observed in Doppler OCT results.

CNN was used to automate these two procedures. ResNet50, Inception, and Xception were used for tissue classification. Among them, Inception presented the best prediction performance with a 99.6% average accuracy rate, while ResNet50 achieved 98.9%, and Xception reached 99.5%. Therefore, our CNN-based endoscopic OCT system demonstrates superior performance compared to the ultrasound-navigation results provided by our control experiments. Slight variations in predictions across different folds were observed, and this phenomenon could be attributed to the physiological differences among the tested kidney samples. Thereafter, Inception was chosen as the final model for tissue recognition. Moreover, we conducted external blind tests using the final Inception model to further reveal the feasibility. Prediction

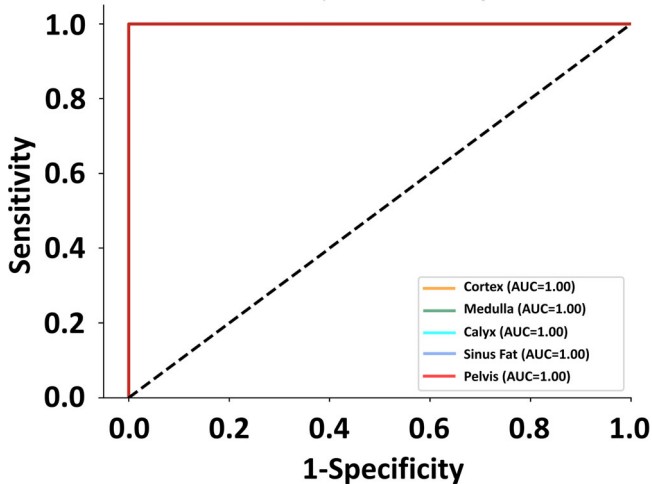

**Fig. 5 | Micro-average ROC curve for five tested kidneys using Inception.**

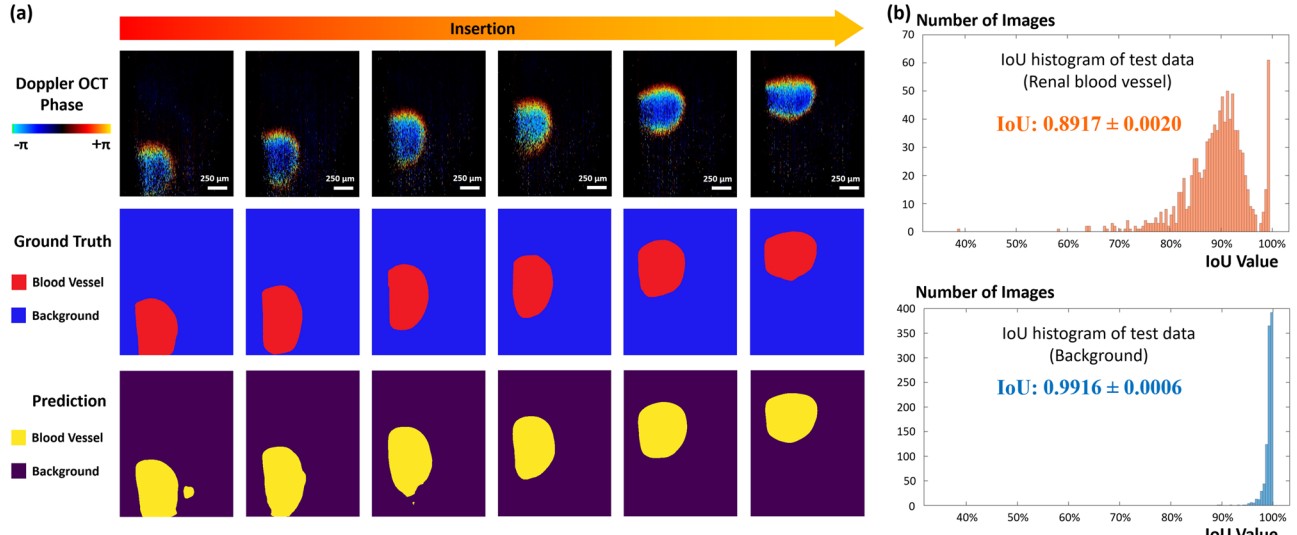

**Fig. 6 | Automatic blood vessel segmentation results. a** Segmentation of renal blood vessel by using nnU-Net. **b** Histograms for IoU of blood vessel and background for the test dataset.

**Table 4 | Blind test tissue prediction results**

| | | Prediction | | | | |
|---|---|---|---|---|---|---|
| | | **Cortex** | **Medulla** | **Calyx** | **Sinus fat** | **Pelvis** |
| **(a) Confusion matrix** | | | | | | |
| Testing Sample | Cortex | **986** | 0 | 0 | 14 | 0 |
| | Medulla | 2 | **998** | 0 | 0 | 0 |
| | Calyx | 0 | 0 | **916** | 84 | 0 |
| | Sinus Fat | 31 | 0 | 0 | **969** | 0 |
| | Pelvis | 0 | 0 | 0 | 0 | **1000** |
| **(b) Accuracy, recall and precision values** | | | | | | |
| Accuracy | | 99.06% | 99.96% | 98.32% | 97.42% | 100.00% |
| Recall | | 98.60% | 99.80% | 91.60% | 96.90% | 100.00% |
| Precision | | 96.95% | 100.00% | 100.00% | 90.82% | 100.00% |

The blod values indicate the correct predictions in the confusion matrices, more directly showing the prediction accucacy.

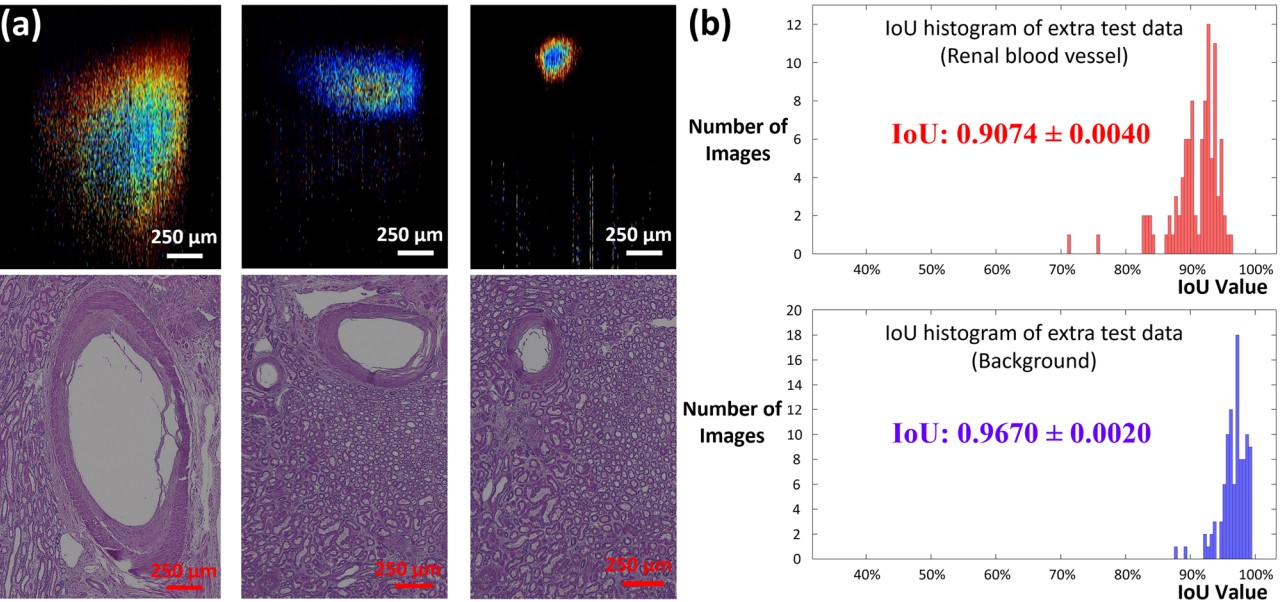

**Fig. 7 | Blind test prediction of blood vessels. a** Doppler OCT images of detected renal blood vessels and the corresponding histology. **b** Histograms for IoU of blood vessels and background for the additional 100-image test dataset.

performances of cortex, medulla, and pelvis were also favorable, but calyx and sinus fat were less effective owing to the substantial fat in the blind test kidney samples. Overall, the CNN-aided renal tissue recognition is promising. In each layer of the Inception model, a range of convolutional filter sizes was incorporated, such as $1 \times 1$, $3 \times 3$, and $5 \times 5$, enabling it to capture information at different scales. This capability is important for distinguishing differences in complex images. Except for detecting simple edges, the convolutional layers can identify and integrate imaging features in sophisticated ways to further improve the classification accuracy. Compared to our previous research using pig models, which achieved an average classification accuracy of 82.6%, the proposed work in this study demonstrates substantially improved performance[20]. Several factors contributed to this enhancement: (1) We further calibrated our system, so it enabled greater imaging quality. Compared to our previous system, the new design achieves ~30% greater imaging depth and improves spatial resolution from 10.6 μm to 5.5 μm. Therefore, it can provide more tissue morphological information in the OCT results; (2) We collected a larger dataset from a greater number of kidney samples. (3) The human kidneys exhibit different OCT imaging characteristics compared to porcine tissues.

An nnU-Net architecture was employed for blood vessel segmentation. The average IoU levels for the test dataset were 0.8917 for blood vessels and 0.9916 for background. External blind tests were conducted to further substantiate the feasibility of blood vessel detection, revealing average IoU levels of 0.9074 for blood flow and 0.9670 for background. These results indicated that this nnU-Net approach was highly effective at detecting blood vessel signals from surrounding tissue in Doppler OCT images.

Since real-time feedback is needed for PCN needle navigation, processing speeds of tissue recognition and blood vessel detection are of great importance. Based on our current computational resources, we have realized tissue recognition prediction in 6.07 ms per image and blood vessel segmentation in 250 ms per image. In practical uses, our probe can perform structural OCT imaging and Doppler OCT imaging simultaneously, thereby enabling both tissue recognition and blood vessel detection functions in real time. In the future, we will further optimize our models and employ enhanced computation resources to improve processing efficiency. For the hardware setup, we plan to use more powerful GPUs and implement multi-GPU configurations to enable parallel training or inference using distributed frameworks. We are currently developing a data pipeline to improve image preprocessing, storage, and data loading. At the same time,

the model is being streamlined to reduce its size and computational complexity.

By utilizing our technique, patients under PCN can expect improved safety and reduced medical costs. Fewer PCN punctures will be required, resulting in reduced tissue and blood vessel injury. The use of CNN prediction can reduce the learning burden and enhance radiologists' working efficiency. Our endoscopic OCT probe system has a user-friendly design with a shallow learning curve. In practical PCN surgeries, radiologists simply insert the endoscope into the inner hollow of the PCN needle, and the imaging results will be automatically displayed, making PCN surgery more convenient to perform. Compared to traditional ultrasound, our technique has demonstrated substantially improved capabilities in both tissue recognition and blood vessel detection. Furthermore, our probe has the potential for broader applications in other surgical guidance, including tumor biopsy, anesthesia needle navigation, localized surgery visualization, drug delivery, etc. We have initiated related studies to explore these applications. For instance, we are investigating its utility in epidural anesthesia for spinal nerve blocks and in localized surgeries involving the extremities.

An important issue in deep learning-based medical imaging research is the informational diversity in large frame datasets. Many imaging datasets include substantial redundancy because imaging protocols are standardized. In this study, our cross-validation results demonstrated consistently high performance in each folder, indicating that our datasets have enough diversity for accurate tissue classification without additional data curation. The stability of the results suggests that our current pipeline is already close to the performance ceiling by the available data. Nevertheless, recent studies with medical imaging datasets containing substantial semantic redundancy demonstrated that carefully selected subsets can match or even exceed the performance of using full training sets. Rajaraman et al. found that using an entropy-based selection of about 55% of the National Institutes of Health Chest X-ray-14 images reduced overfitting and improved both internal and external performance compared to using 100% of the data[35]. Chinn et al. reported similar findings with a customized label-free selection method that eliminates noise and redundancy for the imaging challenges method, which reached near-maximal classification and segmentation performance using 30–60% of retinal OCT and ultrasound images[36]. These studies show that redundant frames can inflate dataset size without adding meaningful information, and redundancy-aware subsampling can improve generalization and reduce computational cost. This is especially relevant to OCT, which provides highly correlated sequential frames. We have recognized its potential value, so we plan to investigate decorrelated and information-based subset selection (e.g., 10%, 25%, 50% of the frames per kidney) combined with kidney-level block bootstrapping, to enable better quantification of the effective sample size of our dataset in our future work.

## Data availability

The data that support this study are available from the repository: https://github.com/thepanlab/Endoscopic_OCT_PCN_guidance

We ensure that our Data Availability Statement complies with the Data Availability policy.

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

## Acknowledgements

This work was supported by grants from the Stephenson Cancer Center (3P30CA225520), Institutional Research Grant number 134128-IRG-19-142-01 from the American Cancer Society, National Science Foundation (OIA-2132161, 2238648), National Institute of Health (R01DK133717), Oklahoma Shared Clinical and Translational Resources (NIGMS U54GM104938), Oklahoma Center for the Advancement of Science and Technology (HR23-071), the medical imaging COBRE (P20 GM135009), and the Midwest Biomedical Accelerator Consortium (MBArC), an NIH Research Evaluation and Commercialization Hub (REACH). Histology service provided by the Tissue Pathology Shared Resource was supported in part by the National Institute of General Medical Sciences COBRE Grant P20GM103639 and National Cancer Institute Grant P30CA225520 of the National Institutes of Health. Financial support was provided by the OU Libraries' Open Access Fund.

## Author contributions

Chen Wang: System design; Experiment; Data analysis; Paper writing and revising. Paul Calle: Deep learning for data processing and analysis; Paper writing and revising. Feng Yan: Experiment; Data analysis. Qinghao Zhang: Experiment; Data analysis. Kar-Ming Fung: Pathology analysis; Paper reviewing. Zhongxin Yu: Pathology analysis; Data analysis. Sean G. Duguay: System development; Paper reviewing. William B. Vanlandingham: System development; Data analysis. Nathan A. Bradley: Pathology analysis; Paper reviewing. Sanjay G. Patel: System development; Paper reviewing. Bradon Nave: Biology data analysis; Paper reviewing. Clint Hostetler: Biology data analysis; Paper reviewing. Ashley Milam: Biology data analysis; Paper reviewing. Chongle Pan: Theoretical development; Paper writing and revising. Qinggong Tang: Theoretical development; Paper writing and revising.

## Competing interests

The authors declare no competing interests.
