## [Transparent Peer Review file · Communications Engineering]

Percutaneous Nephrostomy Guidance by a Convolutional-Neural-Network-Based Optical Coherence Tomography Endoscope

Corresponding Author: Professor Qinggong Tang

Version 0:

Reviewer comments:

Reviewer #2

(Remarks to the Author)

In this manuscript, the team developed and demonstrated an endoscopic optical coherence tomography (OCT) probe for needle navigation with the use of modern machine learning techniques. They have conducted experiments on 31 human kidneys samples to recognize the different tissue types and blood vessels. Their study showed that the renal tissues including cortex, medulla, calyx, sinus fat and pelvis can be effectively classified through the OCT images, and renal blood vessel can be detected through the doppler OCT. The great performance of the proposed techniques is evidenced by the high detection accuracy. Overall, the manuscript is well written and easy to follow with high clinical significance. It will be helpful to the field of optical imaging, surgical intervention. However, the reviewer has several concerns that may require authors' attention to improve their manuscript for publication.

1. More details in the method section regarding the machine learning data processing will be helpful for readers. Especially, what does the intrinsic difference of each tissue type allow the successful tissue classification should be better clarified.
2. OCT scanning and imaging processing speed are also critical for the future real time applications. It appears that the authors only briefly mentioned the imaging processing speed in the last paragraph of the discussion section. The reviewer would suggest the authors to list one table regarding the data scanning and imaging processing speed with their current computation recourse in the results section to highlight the importance of imaging and processing speed for real-time applications.
3. The machine learning technique does show great performance on the tissue classification. It will be helpful if the authors can provide some discussion regarding the effect of the imaging quality (contrast, resolution et al) on the imaging processing method.
4. it will be helpful if the authors can provide some discussion regarding the general use of these proposed OCT and machine learning techniques for the similar applications.
5. The figure 6 clearly showed the comparison of the machine learning recognized vessels verse the ground truth. Would it be possible for the authors show some similar figures for the other tissue types?

Reviewer #3

(Remarks to the Author)

The work offers a promising endoscopic OCT platform with deep learning for tissue recognition and vessel detection. The reported performance (99–100% accuracy) is striking. Given that adjacent OCT frames acquired during a scan are often highly correlated, some portion of the near-perfect metrics may reflect intra-sequence redundancy rather than subject-level generalization. To help clarify and, if feasible, further strengthen the evidence for generalizability, I have a few questions and suggestions below.

- 1) Clarifications on data acquisition and sampling

1.1) Number and independence of acquisition sites/trajectories

Could you please indicate how many unique acquisition sites or scan trajectories per tissue per kidney contributed to the 10,000 training frames?

Were these frames drawn from one or a few continuous scans at a single site, or from multiple spatially separated locations per tissue?

1.2) Spatial/temporal spacing and subsampling

What was the spatial and/or temporal spacing between consecutive frames included in training and testing?

Did you perform any subsampling (e.g., every Nth frame or one frame per B-scan/volume) to mitigate autocorrelation? If so, at what rate; if not, would you consider reporting results under such subsampling?

1.3) Split strategy and potential leakage

In addition to kidney-level splits (as described), were splits also constrained at the acquisition-site/sequence level to prevent near-duplicate or highly overlapping frames from appearing across train/validation/test? If not, it would be helpful to assess this, as sequence-level leakage can inflate apparent performance.

2) Reporting recommendations

2.1) Per-kidney performance with uncertainty

If possible, please provide per-kidney confusion matrices and per-kidney accuracy/recall/precision for both the internal test set (5 kidneys) and the blind test set (5 kidneys), along with 95% confidence intervals (e.g., kidney-level or subject-level bootstrapping).

The current presentation focuses on pooled internal and pooled blind-test metrics; per-kidney variability with 95% CIs would help readers gauge robustness across subjects.

2.2) Inter-kidney error patterns

It would also be informative to examine inter-kidney classification behavior (e.g., whether certain kidneys or tissue types systematically drive errors) and whether models trained on one subset of kidneys generalize consistently to others, reporting 95% CIs where applicable.

3) Suggested robustness analyses and ablations

3.1) Frame de-correlation ablation

Consider training and evaluating with heavy temporal/spatial subsampling (e.g., every Nth B-scan/frame; one representative slice per 3D block) to reduce adjacent-frame redundancy. Reporting the performance change (with 95% CIs) relative to full-frame training would clarify sensitivity to autocorrelation.

3.2) As an “effective sample size” control, you might also train with 10%, 25%, and 50% of de-correlated frames to assess data efficiency and potential reliance on redundancy. Please report kidney-level 95% CIs (e.g., block/kidney bootstrap).

3.3) Sequence-level isolation

Enforce that entire scan sequences/trajectories (not just kidneys) are confined to a single split, and compare results against kidney-only splits. Quantify any differences with 95% CIs.

3.4) Leave-one-kidney-out cross-validation

If feasible, a strict leave-one-kidney-out CV would directly assess subject-level generalization and clarify the “17-fold” phrasing versus the implemented scheme. Please report per-fold/per-kidney metrics with 95% CIs.

3.5) Loss functions and class balancing

Explore cross-entropy vs. focal loss and class-weighted CE (particularly for more challenging classes such as calyx/sinus fat), and report effects on per-class and per-kidney recall/precision with 95% CIs.

3.6) Randomized-label control

A shuffled-label training control should show the expected collapse in performance; please quantify with 95% CIs to bound any residual signal.

Input ablation and attribution

Occlusion/erasure tests or attribution maps could help verify that decisions are driven by tissue structure rather than acquisition artifacts. Where metrics are computed, include 95% CIs.

3.7) Effective sample size and interval re-estimation

If frame autocorrelation is substantial, consider estimating an effective sample size and re-computing 95% CIs accordingly to provide more conservative uncertainty bounds.

Collectively, these clarifications and analyses—especially per/inter-kidney reporting with 95% confidence intervals, sequence-level isolation, de-correlation ablations, and leave-one-kidney-out CV—would substantially bolster the case that the excellent pooled metrics primarily reflect genuine across-kidney generalization rather than redundancy among adjacent OCT frames or narrow acquisition conditions. If some of these suggestions are not feasible at this time, even partial analyses or sensitivity checks (with 95% CIs) would still be very informative.

Version 1:

Reviewer comments:

Reviewer #2

(Remarks to the Author)

The authors have addressed all of the concerns raised by the reviewer and it may be publishable at the current format.

Reviewer #3

(Remarks to the Author)

The authors addressed all my comments, manuscript can be accepted for publication.

Dear reviewers,

We hereby submit our revised manuscript “Percutaneous Nephrostomy Guidance by a Convolutional-Neural-Network-Based Optical Coherence Tomography Endoscope” for consideration for publication in *Communications Engineering*. We appreciate all the reviewers for your largely positive, highly constructive comments and suggestions. We have made the necessary revisions and emendations along the lines recommended by them. We believe that the manuscript has improved significantly. The changes are listed in red font in the revised manuscript.

Here in this author response letter, our responses to the reviewers’ comments (in black) are listed in blue font as follows:

Reviewer #2:

In this manuscript, the team developed and demonstrated an endoscopic optical coherence tomography (OCT) probe for needle navigation with the use of modern machine learning techniques. They have conducted experiments on 31 human kidneys samples to recognize the different tissue types and blood vessels. Their study showed that the renal tissues including cortex, medulla, calyx, sinus fat and pelvis can be effectively classified through the OCT images, and renal blood vessel can be detected through the doppler OCT. The great performance of the proposed techniques is evidenced by the high detection accuracy. Overall, the manuscript is well written and easy to follow with high clinical significance. It will be helpful to the field of optical imaging, surgical intervention. However, the reviewer has several concerns that may require authors' attention to improve their manuscript for publication.

Thanks for your comments.

1. More details in the method section regarding the machine learning data processing will be helpful for readers. Especially, what does the intrinsic difference of each tissue type allow the successful tissue classification should be better clarified.

Thanks for the suggestion. We added descriptions of the different tissue types in terms of their locations and histological structures. We added ‘Each of the five tissue layers is characterized by a specific anatomical location and histological structure. Cortex is located at the outermost layer of the kidney and it is composed of renal corpuscles, tubules and cortical interstitial tissue. The renal medulla, located beneath the outer cortex, consists primarily of loops of Henle, collecting ducts and vasa recta. Renal calyx is located at the innermost of the kidneys surrounding the renal pyramids formed by medulla tissue, and it is composed of transitional epithelium and smooth muscle. Renal sinus fat is situated between adjacent renal pyramids and comprises adipocytes and connective tissues. Pelvis is positioned toward the central medial side of the kidney and serves as the cavity that collects the urine from the calyces, so it appears as an empty or fluid-filled space during imaging.’ in Section ‘Data Acquisition of Human Kidneys and CNN-based Prediction’ and ‘To improve working efficiency and reduce the working burden of doctors, we propose to use deep learning methods to automate the tissue recognition and blood vessel detection procedures.’ in Section ‘CNN-Aided Classification Task, Segmentation Task, and Blind Test’.

2. OCT scanning and imaging processing speed are also critical for the future real time applications. It appears that the authors only briefly mentioned the imaging processing speed in the last paragraph of the discussion section. The reviewer would suggest the authors to list one table regarding the data scanning and imaging processing speed with their current computation recourse in the results section to highlight the importance of imaging and processing speed for real-time applications.

Thanks for the suggestion. We have added the description and a table of GPU specs and prediction speeds in the results

section as shown below:

“To realize real-time feedback during PCN navigation, a high-performance server was employed to accelerate data processing within the CNN framework. In this study, we used a Linux server with Ubuntu 20.04.6 LTS with two NVIDIA RTX A6000 GPUs for all CNN tasks. For tissue recognition using a batch size of 32 images, it required an average prediction time of 6.07 ms per image. For blood vessel segmentation, the average processing time was 250 ms. These results affirmed the real-time performance of our OCT imaging platform, illustrating its suitability for clinical applications where timely decision-making is critical. By utilizing our technique to reduce tissue damage and blood vessel injury, patients under PCN can expect improved safety and reduced medical costs. Basic information of the GPU and prediction performance are listed in Table I.”

TABLE I. Data processing hardware and prediction performance

GPU Information					Prediction Performance	
GPU model	Memory	Memory Interface	Memory Bandwidth	CUDA cores	Tissue Recognition Time/per image	Blood Vessel Segmentation Time/per image
NVIDIA RTX A6000	48 GB	384-bit	768 GB/s	10,752	6.07 ms	250 ms

We also added further discussion in Discussion section as below:

“Since real-time feedback is needed for PCN needle navigation, processing speeds of tissue recognition and blood vessel detection are of great importance. Based on our current computation resource, we have realized tissue recognition prediction in 6.07 ms per image and blood vessel segmentation in 250 ms per image. In the future, we will further optimize our models and employ enhanced computation resources to improve processing efficiency. For the hardware setup, we plan to use more powerful GPUs and implement multi-GPU configurations to enable parallel training or inference using distributed frameworks. We are currently developing a data pipeline to improve image preprocessing, storage, and data loading. At the same time, the model is being streamlined to reduce its size and computational complexity.”

3. The machine learning technique does show great performance on the tissue classification. It will be helpful if the authors can provide some discussion regarding the effect of the imaging quality (contrast, resolution et al) on the imaging processing method.

Thanks for the comment. We further compared our current results with our previous ones. We added the following content in Discussion section:

“Compared to our previous research using pig models which achieved an average classification accuracy of 82.6%, the proposed work in this study demonstrates substantially improved performance. Several factors contributed to this enhancement: 1) We further calibrated our system, so it enabled greater imaging quality. Compared to our previous system, the new design achieves ~30% greater imaging depth and improves spatial resolution from 10.6 μm to 5.5 μm . Therefore, it can provide more tissue morphological information in the OCT results; 2) We collected a larger dataset from a greater number of kidney samples. 3) The human kidneys exhibit different OCT imaging characteristics compared to porcine tissues.”

4. It will be helpful if the authors can provide some discussion regarding the general use of these proposed OCT and machine learning techniques for the similar applications?

Thanks for the comment. We have added the following content in Discussion:

“By utilizing our technique, patients under PCN can expect improved safety and reduced medical costs. Fewer PCN punctures will be required, resulting in reduced tissue and blood vessel injury. The use of CNN prediction can reduce the learning burden and enhance radiologists’ working efficiency. Our endoscopic OCT probe system has a user-friendly design with a shallow learning curve. In practical PCN surgeries, radiologists simply insert the endoscope into the inner hollow of PCN needle, and the imaging results will be automatically displayed, making PCN surgery more convenient to perform. Compared to traditional ultrasound, our technique has demonstrated substantially improved capabilities in both tissue recognition and blood vessel detection. Furthermore, our probe has the potential for broader applications in other surgical guidance, including tumor biopsy, anesthesia needle navigation, localized surgery visualization, drug delivery etc. We have initiated related studies to explore these applications. For instance, we are investigating its utility in epidural anesthesia for spinal nerve blocks and in localized surgeries involving the extremities.”

5. The figure 6 clearly showed the comparison of the machine learning recognized vessels verse the ground truth. Would it be possible for the authors show some similar figures for the other tissue types??

Thanks for the comments. Since the Doppler OCT only detects the flow information, only blood flow area can be displayed in the Doppler OCT results. Also, the aim of using Doppler OCT is only to detect blood vessel areas. To show other tissue types we can rely on the structural OCT results which can recognize different tissue types. It is noteworthy that, based on the high-speed data processing, structural OCT imaging and Doppler OCT imaging can be performed simultaneously in real time. Therefore, we can observe the tissue type and blood vessel at the same time during PCN procedure navigated by our system. We have added the following sentence in Discussion:

“In practical uses, our probe can perform structural OCT imaging and Doppler OCT imaging simultaneously, thereby enabling both tissue recognition and blood vessel detection functions in real time.”

Reviewer #3:

The work offers a promising endoscopic OCT platform with deep learning for tissue recognition and vessel detection. The reported performance (99–100% accuracy) is striking. Given that adjacent OCT frames acquired during a scan are often highly correlated, some portion of the near-perfect metrics may reflect intra-sequence redundancy rather than subject-level generalization. To help clarify and, if feasible, further strengthen the evidence for generalizability, I have a few questions and suggestions below:

Thanks for your comments.

1) Clarifications on data acquisition and sampling

1.1) Number and independence of acquisition sites/trajectories

Could you please indicate how many unique acquisition sites or scan trajectories per tissue per kidney contributed to the 10,000 training frames?

Were these frames drawn from one or a few continuous scans at a single site, or from multiple spatially separated locations per tissue?

Thanks for your comments. For each tissue type in each kidney sample, we randomly selected over 60 sites initially and started the scanning on these sites. For each site, we conducted around 200 adjacent cross-sectional imaging results. As a result, we successfully obtained over 10,000 OCT images from each tissue type in each kidney sample. The 10,000 results were all obtained from different locations, not from one or a few continuous scans at a single site. After selecting the site, we set a specific distance parameter between adjacent cross-sectional scanning of the ~200

results, and then the system could complete the scanning automatically. More details are shown in the following figure which has been added in the Figure S1 of the revised supplementary data:

1.2) Spatial/temporal spacing and subsampling

What was the spatial and/or temporal spacing between consecutive frames included in training and testing?

Did you perform any subsampling (e.g., every Nth frame or one frame per B-scan/volume) to mitigate autocorrelation? If so, at what rate; if not, would you consider reporting results under such subsampling?

Thanks for your comments. In our scanning, each frame was obtained independently (as shown in comment 1.1), so we will not consider reporting results under subsampling.

1.3) Split strategy and potential leakage

In addition to kidney-level splits (as described), were splits also constrained at the acquisition-site/sequence level to prevent near-duplicate or highly overlapping frames from appearing across train/validation/test? If not, it would be helpful to assess this, as sequence-level leakage can inflate apparent performance.

Thanks for your comments. When we selected the scanning sites we distributed them in all available areas of the whole kidneys. For instance, when we scanned renal cortex, we selected the sites in apical, upper, middle, posterior, and lower areas to make our data comprehensive and persuasive. Importantly, our data partitioning was performed strictly at the *kidney level*: all image sequences (i.e., all frames from all scanning sites within a given kidney) are assigned exclusively to a single split—training, validation, or testing. As a result, no acquisition site, scan sequence, or contiguous frame set from the same kidney can appear across multiple splits, fully preventing sequence-level or frame-level leakage. Therefore, highly overlapping or near-duplicate frames cannot occur across partitions.

2) Reporting recommendations

2.1) Per-kidney performance with uncertainty

If possible, please provide per-kidney confusion matrices and per-kidney accuracy/recall/precision for both the internal test set (5 kidneys) and the blind test set (5 kidneys), along with 95% confidence intervals (e.g., kidney-level or subject-level bootstrapping).

The current presentation focuses on pooled internal and pooled blind-test metrics; per-kidney variability with 95% CIs would help readers gauge robustness across subjects.

Thanks for your comments. The matrices for the classification internal results are shown below and have been added

in the revised supplementary document:

Internal kidney - 1

		Predicted	Predicted	Predicted	Predicted	Predicted
		Cortex	Medulla	Calyx	Fat	Pelvis
Truth	Cortex	10000	0	0	0	0
Truth	Medulla	0	10000	0	0	0
Truth	Calyx	0	0	10000	0	0
Truth	Fat	0	0	0	10000	0
Truth	Pelvis	0	0	0	0	10000

Internal kidney - 2

		Predicted	Predicted	Predicted	Predicted	Predicted
		Cortex	Medulla	Calyx	Fat	Pelvis
Truth	Cortex	9947	0	53	0	0
Truth	Medulla	0	10000	0	0	0
Truth	Calyx	0	0	10000	0	0
Truth	Fat	47	0	54	9899	0
Truth	Pelvis	0	0	0	0	10000

Internal kidney - 3

		Predicted	Predicted	Predicted	Predicted	Predicted
		Cortex	Medulla	Calyx	Fat	Pelvis
Truth	Cortex	10000	0	0	0	0
Truth	Medulla	0	10000	0	0	0
Truth	Calyx	0	0	10000	0	0
Truth	Fat	0	0	0	10000	0
Truth	Pelvis	0	0	0	0	10000

Internal kidney - 4

		Predicted	Predicted	Predicted	Predicted	Predicted
		Cortex	Medulla	Calyx	Fat	Pelvis
Truth	Cortex	10000	0	0	0	0
Truth	Medulla	0	9990	10	0	0
Truth	Calyx	0	0	10000	0	0
Truth	Fat	0	0	0	10000	0
Truth	Pelvis	0	0	0	0	10000

Internal kidney - 5

		Predicted	Predicted	Predicted	Predicted	Predicted
		Cortex	Medulla	Calyx	Fat	Pelvis
Truth	Cortex	10000	0	0	0	0
Truth	Medulla	0	10000	0	0	0
Truth	Calyx	0	0	9999	1	0
Truth	Fat	0	0	0	10000	0
Truth	Pelvis	0	0	0	0	10000

For blind test, it can be seen in Table 5 of the manuscript.

2.2) Inter-kidney error patterns

It would also be informative to examine inter-kidney classification behavior (e.g., whether certain kidneys or tissue types systematically drive errors) and whether models trained on one subset of kidneys generalize consistently to others, reporting 95% CIs where applicable.

Thanks for your comments. Here below are the results to evaluate inter-kidney classification behavior and we have added them in the supplementary document.

Per-kidney analysis:

	Accuracy	CI lower	CI upper
Internal kidney - 1	1.0000	0.999923176725854	0.9999999999999999
Internal kidney - 2	1.0000	0.999923176725854	0.9999999999999999
Internal kidney - 3	1.0000	0.999923176725854	0.9999999999999999
Internal kidney - 4	0.9998	0.999631850671226	0.999891356783937
Internal kidney - 5	1.0000	0.999886710298608	0.999996469500177

Per-tissue analysis:

Precision

	calyx			cortex			fat			medulla			pelvis		
	value	lower CI	upper CI	value	lower CI	upper CI	value	lower CI	upper CI	value	lower CI	upper CI	value	lower CI	upper CI
Internal kidney - 1	1.0000	0.9996	1.0000	1.0000	0.9996	1.0000	1.0000	0.9996	1.0000	1.0000	0.9996	1.0000	1.0000	0.9996	1.0000
Internal kidney - 2	0.9894	0.9872	0.9912	0.9953	0.9938	0.9965	1.0000	0.9996	1.0000	1.0000	0.9996	1.0000	1.0000	0.9996	1.0000
Internal kidney - 3	1.0000	0.9996	1.0000	1.0000	0.9996	1.0000	1.0000	0.9996	1.0000	1.0000	0.9996	1.0000	1.0000	0.9996	1.0000
Internal kidney - 4	0.9990	0.9982	0.9995	1.0000	0.9996	1.0000	1.0000	0.9996	1.0000	1.0000	0.9996	1.0000	1.0000	0.9996	1.0000
Internal kidney - 5	1.0000	0.9996	1.0000	1.0000	0.9996	1.0000	0.9999	0.9994	1.0000	1.0000	0.9996	1.0000	1.0000	0.9996	1.0000

Recall

	calyx			cortex			fat			medulla			pelvis		
	value	lower CI	upper CI	value	lower CI	upper CI	value	lower CI	upper CI	value	lower CI	upper CI	value	lower CI	upper CI
Internal kidney - 1	1.0000	0.9996	1.0000	1.0000	0.9996	1.0000	1.0000	0.9996	1.0000	1.0000	0.9996	1.0000	1.0000	0.9996	1.0000
Internal kidney - 2	1.0000	0.9996	1.0000	0.9947	0.9931	0.9959	0.9899	0.9877	0.9917	1.0000	0.9996	1.0000	1.0000	0.9996	1.0000
Internal kidney - 3	1.0000	0.9996	1.0000	1.0000	0.9996	1.0000	1.0000	0.9996	1.0000	1.0000	0.9996	1.0000	1.0000	0.9996	1.0000
Internal kidney - 4	1.0000	0.9996	1.0000	1.0000	0.9996	1.0000	1.0000	0.9996	1.0000	0.9990	0.9982	0.9995	1.0000	0.9996	1.0000
Internal kidney - 5	0.9999	0.9994	1.0000	1.0000	0.9996	1.0000	1.0000	0.9996	1.0000	1.0000	0.9996	1.0000	1.0000	0.9996	1.0000

3) Suggested robustness analyses and ablations

3.1) Frame de-correlation ablation

Consider training and evaluating with heavy temporal/spatial subsampling (e.g., every Nth B-scan/frame; one representative slice per 3D block) to reduce adjacent-frame redundancy. Reporting the performance change (with 95% CIs) relative to full-frame training would clarify sensitivity to autocorrelation.

Thanks for your comments. For classification, the test is based on a separate kidney so there is no room for spatial subsampling.

3.2) As an “effective sample size” control, you might also train with 10%, 25%, and 50% of de-correlated frames to assess data efficiency and potential reliance on redundancy. Please report kidney-level 95% CIs (e.g., block/kidney bootstrap).

Thanks for your comments. We appreciate the reviewer’s suggestion regarding effective sample size and redundancy. Our current models have realized to be **high-performance and stable across folds in kidney-level**, indicating that the dataset as used in this study contains sufficient diversity for reliable generalization. Therefore, we did not perform additional training runs at 10%, 25%, and 50% of the de-correlated frames for this submission.

However, we fully agree that this line of analysis is valuable and we will incorporate it into our **future work**. Recent studies have shown that large medical imaging datasets often contain substantial semantic redundancy. For instance, Rajaraman et al. demonstrated that a model trained on only ~55% of “informative” chest X-ray images outperformed a model trained on the full dataset. Chinn et al. (ENRICH) showed that carefully selected subsets comprising 30–60% of ultrasound and OCT images can achieve performance comparable to training on the full dataset. These findings indicate that, for many medical imaging datasets, reducing redundancy can preserve or improve performance while significantly increasing training efficiency.

Motivated by these results, we will investigate **subsampling and de-correlation strategies** to quantify the effective sample size in our kidney dataset and to determine whether a curated subset can match or even improve upon our current performance. We have added the following content in the revised Discussion section.

An important issue in deep learning-based medical imaging research is the informational diversity in large frame datasets. Many imaging datasets include substantial redundancy because imaging protocols are standardized. In this study, our cross-validation results demonstrated consistently high performance in each folder, indicating that our datasets have enough diversity for accurate tissue classification without additional data curation. The stability of the results suggests that our current pipeline is already close to the performance ceiling by the available data. Nevertheless, recent studies with medical imaging datasets containing substantial semantic redundancy demonstrated that carefully selected subsets can match or even exceed the performance using full training sets. Rajaraman et al. found that using an entropy-based selection of about 55% of the National Institutes of Health Chest X-ray-14 images reduced overfitting and improved both internal and external performance compared to using 100% of the data [1]. Chinn et al. reported similar findings with a customized label-free selection eliminate noise and redundancy for imaging challenges method, which reached near-maximal classification and segmentation performance using 30-60% of retinal OCT and ultrasound images[2]. These studies show that redundant frames can inflate dataset size without adding meaningful information, and redundancy-aware subsampling can improve generalization and reduce computational cost. This is especially relevant to OCT which provides highly correlated sequential frames. We have recognized its potential value, so we plan to investigate de-correlated and information-based subset selection (e.g., 10%, 25%, 50% of the frames per kidney) combined with kidney-level block bootstrapping, to enable better quantification of the effective sample size of our dataset in our future work.

3.3) Sequence-level isolation

Enforce that entire scan sequences/trajectories (not just kidneys) are confined to a single split, and compare results against kidney-only splits. Quantify any differences with 95% CIs.

We thank the reviewer for raising this important point about potential leakage across scan trajectories. In our dataset, the hierarchy is: kidney → scan sequence/trajectory → frames. For all experiments in this work, cross-validation splits are defined at the kidney level. All scan sequences and frames belonging to a given kidney are assigned to the same fold. In other words, there is no mixing of sequences from the same kidney across training, validation, or test sets.

Under this design, “sequence-level isolation” guarantees kidney-level isolation. Applying sequence-level isolation would result in the same data partitions as the existing kidney-level splits, leading to identical performance estimates and confidence intervals. To prevent confusion, we have clarified this in the Method:

Our dataset follows the hierarchy of: kidney → scan sequence/trajectory → frames.

3.4) Leave-one-kidney-out cross-validation

If feasible, a strict leave-one-kidney-out CV would directly assess subject-level generalization and clarify the “17-fold” phrasing versus the implemented scheme. Please report per-fold/per-kidney metrics with 95% CIs.

Thanks for your comments. We agree that strict leave-one-kidney-out cross-validation (LOKO-CV) would directly assess subject-level generalization. However, implementing full LOKO-CV would require training 17 completely independent models, each with full training + validation + hyperparameter optimization pipeline. For practical uses, the 17 independent projects would need a time and computational resource requirement that is not practical within a feasible research framework. Importantly, our existing scheme already enforces strict kidney-level isolation: all frames and sequences from each kidney are confined to a single fold. Thus, the current 17-fold setup already evaluates across-kidney generalization while remaining computationally tractable. We have clarified the “17-fold” phrasing accordingly. Full LOKO-CV is a valuable direction, and we plan to conduct this in our future work.

3.5) Loss functions and class balancing

Explore cross-entropy vs. focal loss and class-weighted CE (particularly for more challenging classes such as calyx/sinus fat), and report effects on per-class and per-kidney recall/precision with 95% CIs.

Thanks for your comments. We appreciate the reviewer’s suggestion to evaluate focal loss and class-weighted cross-entropy. These approaches are typically helpful when models struggle with minority or challenging classes. In our case, however, the per-class results already show consistently high recall and precision across all tissue categories, including the more challenging classes (e.g., calyx and sinus fat). This indicates that our current model is not biased toward majority classes and is already handling class imbalance effectively.

Given the uniformly strong per-class and per-kidney performance, we expect alternative loss formulations to yield only marginal improvements while substantially increasing training cost. For these reasons, we have not incorporated additional loss-function experiments in this revision, but we agree that evaluating focal loss or class-weighted CE would be a valuable direction for future work, particularly in scenarios with more severe imbalance.

3.6) Randomized-label control

A shuffled-label training control should show the expected collapse in performance; please quantify with 95% CIs to bound any residual signal.

Input ablation and attribution

Occlusion/erasure tests or attribution maps could help verify that decisions are driven by tissue structure rather than acquisition artifacts. Where metrics are computed, include 95% CIs.

Thanks for your comments. We performed a randomized-label control on one validation subject, in which all tissue labels were shuffled while the images remained unchanged. As shown in the learning curves, the model fails to learn any meaningful structure under this condition

The resulting confusion matrix is:

		Predicted				
		Cortex	Medulla	Calyx	Sinus Fat	Pelvis
Truth	Cortex	2617	766	775	4418	1424
	Medulla	2662	735	751	4422	1430
	Calyx	2694	726	723	4484	1373
	Sinus Fat	2714	672	758	4436	1420
	Pelvis	2551	761	774	4422	1472

Class Cortex

Precision: 0.1977 (95% CI: 0.1910 to 0.2046)

Recall: 0.2617 (95% CI: 0.2532 to 0.2704)

Class Medulla:

Precision: 0.2008 (95% CI: 0.1882 to 0.2141)

Recall: 0.0735 (95% CI: 0.0685 to 0.0788)

Class Calyx:

Precision: 0.1912 (95% CI: 0.1790 to 0.2041)

Recall: 0.0723 (95% CI: 0.0674 to 0.0775)

Class Sinut Fat:

Precision: 0.1998 (95% CI: 0.1946 to 0.2051)

Recall: 0.4436 (95% CI: 0.4339 to 0.4534)

Class Pelvis:

Precision: 0.2068 (95% CI: 0.1975 to 0.2163)

Recall: 0.1472 (95% CI: 0.1404 to 0.1543)

These values cluster around ~20% precision, consistent with a 5-class random classifier. The uniformly poor and near-chance performance presents that the network is unable to extract any discriminative signal when label-image correspondence is removed.

This randomized-label collapse demonstrates that our observed high performance in the main experiments is not driven by acquisition artifacts, data leakage, or spurious correlations. Instead, it reflects genuine learning of tissue-specific structural features.

Regarding occlusion/erasure tests, we agree that these techniques can provide additional qualitative insight into model decision pathways. However, implementing full ablation experiments across all folds would require substantial additional computational resource beyond the scope of the current study. Importantly, our results already provide strong evidence that decisions are driven by meaningful tissue structure: 1) all classes achieve uniformly high and stable recall/precision across kidneys, 2) there is no performance inflation under randomized labels, and 3) training/validation/test sets are fully isolated at the kidney level, minimizing the risk of artifact-driven leakage.

3.7) Effective sample size and interval re-estimation

If frame autocorrelation is substantial, consider estimating an effective sample size and re-computing 95% CIs accordingly to provide more conservative uncertainty bounds.

Thanks for your comments. We appreciate the reviewer's point regarding frame autocorrelation and its potential effect on uncertainty estimates. In our study, however, performance metrics and confidence intervals are computed strictly at the kidney level, and each fold's test set consists of an entirely different kidney from those used in training and validation. Since no frames or sequences from a given kidney appear in more than one split, within-kidney frame autocorrelation does not propagate into our uncertainty estimates.

Again, we would like to thank all the reviewers for your time and efforts.

Sincerely yours,

Qinggong Tang, Ph.D.

Associate Professor of Stephenson School of Biomedical Engineering (SBME)

Endowed Stephenson Professor

Director, Medical Imaging Technology Development Core, NIH P20 COBRE

Chair, Translation Simulation Committee, SBME

Associate Member – Stephenson Cancer Center

University of Oklahoma

407 Gallogly Hall

173 Felgar St, Norman, OK 73019

qtang@ou.edu | Google Scholar

Website: <http://tanglab.oucreate.com/>

Twitter: @TangLab2

Reference

1. S. Rajaraman, G. Zamzmi, F. Yang, Z. Liang, Z. Xue, and S. Antani, "Semantically redundant training data removal and deep model classification performance: A study with chest X-rays," *Computerized Medical Imaging and Graphics* **115**, 102379 (2024).
2. E. Chinn, R. Arora, R. Arnaout, and R. Arnaout, "ENRICHing medical imaging training sets enables more efficient machine learning," *J Am Med Inform Assoc* **30**, 1079-1090 (2023).